## [Peer Review File · EMBO Reports]

p53 status determines the epigenetic response to demethylating agents Azacitidine and Decitabine.

Emma Hands, Arndt Wallmann, Gabrielle Oxley, Sophie Storrar, Rochelle D'Souza, and Mathew Van de Pette

Corresponding author(s): Mathew Van de Pette (mathew.vandepette@ukhsa.gov.uk)

Review Timeline:

Submission Date:	26th Mar 25
Editorial Decision:	28th Apr 25
Revision Received:	21st Aug 25
Editorial Decision:	3rd Oct 25
Revision Received:	1st Dec 25
Accepted:	8th Dec 25

Editor: Achim Breiling

Transaction Report:

Dear Dr. Van de Pette,

Thank you for the transfer of your manuscript to EMBO reports. I have now received the reports from the three referees that were asked to evaluate your study, which can be found at the end of this email.

As you will see, the referees think that these findings are of interest. However, they have several comments, concerns, and suggestions, indicating that a major revision of the manuscript is necessary to allow publication of the study in EMBO reports. As the reports are below, and all the referee concerns need to be addressed, I will not detail them here.

Given the constructive referee comments, I would like to invite you to revise your manuscript with the understanding that the concerns of the referees must be addressed in the revised manuscript and in a detailed point-by-point response. Acceptance of your manuscript will depend on a positive outcome of a second round of review. It is EMBO reports policy to allow a single round of revision only and acceptance of the manuscript will therefore depend on the completeness of your responses included in the next, final version of the manuscript.

- 1) a .docx formatted version of the final manuscript text (including legends for main figures, EV figures and tables), but without the figures included. Figure legends should be compiled at the end of the manuscript text.
- 2) individual production quality figure files as .eps, .tif, .jpg (one file per figure), of main figures and EV figures. Please upload these as separate, individual files upon re-submission.

- 4) a complete author checklist, which you can download from our author guidelines (<https://www.embopress.org/page/journal/14693178/authorguide>). Please insert page numbers in the checklist to indicate where the requested information can be found in the manuscript. The completed author checklist will also be part of the RPF.

- 5) that primary datasets produced in this study (e.g. RNA-seq, ChIP-seq, structural and array data) are deposited in an

appropriate public database. If no primary datasets have been deposited, please also state this in a dedicated section (e.g. 'No primary datasets have been generated and deposited'), see below.

The accession numbers and database should be listed in a formal "Data Availability" section that follows the model below. This is now mandatory (like the COI statement). Please note that the Data Availability Section is restricted to new primary data that are part of this study. This section is mandatory. As indicated above, if no primary datasets have been deposited, please state this in this section

Data availability

6) We now request the publication of original source data with the aim of making primary data more accessible and transparent to the reader. You will receive a separate email with instructions for providing source data with your revised manuscript, including information how to upload and organize the files.

8) Regarding data quantification and statistics, please make sure that the number "n" for how many independent experiments were performed, their nature (biological versus technical replicates), the bars and error bars (e.g. SEM, SD) and the test used to calculate p-values is indicated in the respective figure legends (also for EV and Appendix figures). Please also check that all the p-values are explained in the legend, and that these fit to those shown in the figure. Please provide statistical testing where applicable. Please avoid the phrase 'independent experiment', but clearly state if these were biological or technical replicates. Please also indicate (e.g. with n.s.) if testing was performed, but the differences are not significant. In case n=2, please show the data as separate datapoints without error bars and statistics. See also: <http://www.embopress.org/page/journal/14693178/authorguide#statisticalanalysis>

9) Please add scale bars of similar style and thickness to microscopic images, using clearly visible black or white bars (depending on the background). Please place these in the lower right corner of the images themselves. Please do not write on or near the bars in the image but define the size in the respective figure legend.

10) Please also note our reference format:

12) We now use CRedit to specify the contributions of each author in the journal submission system. CRedit replaces the author contribution section. Please use the free text box to provide more detailed descriptions and do NOT provide your final manuscript text file with an author contributions section. See also our guide to authors: <https://www.embopress.org/page/journal/14693178/authorguide#authorshipguidelines>

13) All Materials and Methods need to be described in the main text using our 'Structured Methods' format, which is required for

all research articles. According to this format, the Methods section should include a Reagents and Tools Table (listing key reagents, experimental models, software, and relevant equipment and including their sources and relevant identifiers), uploaded as separate file, and a Methods section in which we encourage the authors to describe their methods using a step-by-step protocol format with bullet points, to facilitate the adoption of the methodologies across labs. More information on how to adhere to this format as well as downloadable templates (.doc) for the Reagents and Tools Table can be found in our author guidelines (section 'Structured Methods'):

Please move all tables with primer or antibody information from the Methods section to the Reagents and Tools Table.

14) Please add up to five keywords to the manuscript and order the sections like this, using (only) these names: Title page - Abstract - Keywords - Introduction - Results - Discussion - Methods - Data availability section - Acknowledgements - Disclosure and Competing Interests Statement - References - Figure legends - Expanded View Figure legends

Please include the funding information into the Acknowledgements section.

15) Please make sure that all the funding information is also entered into the online submission system and that it is complete and similar to the one in the acknowledgement section of the manuscript text file.

Please note that corresponding authors are required to supply an ORCID ID upon submission of a revised manuscript and an institutional e-mail address. Please make sure that the corresponding author provides an ORCID in the submission system. Please find instructions on how to link the ORCID ID to the account in our manuscript tracking system in our author guidelines: <http://www.embopress.org/page/journal/14693178/authorguide#authorshipguidelines>

I look forward to seeing a revised form of your manuscript when it is ready.

Yours sincerely

Referee #1:

1. Does this manuscript report a single key finding? YES

The loss of p53 changes the epigenetic landscape after Aza/Dac treatment.

2. Is the reported work of significance (YES), or does it describe a confirmatory finding or one that has already been documented using other methods or in other organisms etc (NO)? YES

Further expands how cells respond to changes in methylation but identifies a new possible biomarker for p53 status.

3. Is it of general interest to the molecular biology community? YES

Due to a deeper look into the further effects of Aza and Dac on p53 and the epigenetic landscape.

4. Is the single major finding robustly documented using independent lines of experimental evidence (YES), or is it really just a preliminary report requiring significant further data to become convincing, and thus more suited to a longer-format article (NO)? YES/NO

Yes, No

The manuscript presented by Hands et al., uses Aza and Dac which are DNA methyltransferase inhibitors and looks at how, when given to cells at sub toxic concentrations changes in the methylation status of promoters occurs, this impacts the transcriptional profiles of treated cells. They also observed that R-loops were primarily formed in promoter regions when cells were treated with Aza vs the termination sites when treated with Dac. They also found out that knocking out p53 caused an increased amount of R-loop staining after Aza and Dac treatment when compared to the untreated controls. The authors used a p53 inducible cell line (H1299) to show that when p53 was induced R-loop staining intensity was depleted even after treatment with Aza and Dac when compared to the non-induced cell line. The main takeaway was the fact that restoration of p53 is able to stabilise the epigenetic landscape and restore the R-loops. A strength of this manuscript lies in the discussion where a paragraph is set aside which mentions some limitations of the study, for example how variable the S9.6 antibody is, although this raises a question if another antibody should be used instead of the S9.6 antibody as this antibody has been shown to bind to dsRNA as well as the RNA-DNA hybrids.

The major claims are that p53 affects the formation of R-loops upon treatment and that p53 is a key player in how the cell responds to Aza and Dac treatment. The findings are significant as there has not been any previous work linking Aza and Dac treatment to p53 and its activation. Although previous work in the field has shown the dependence of p53 on R-loop formation in other areas such as during HPV (<https://www.pnas.org/doi/10.1073/pnas.2305907120>). Another major claim was that Aza and Dac treatment causes global remapping of R-loops, this work is significant because Aza and Dac are currently used in the treatment of AML and MDS so further research into how these drugs affect 'normal' cells in the body during treatment is important, to prevent any off-shoot effects occurring. Although the S9.6 antibody can be quite variable, as described earlier, Hands et al., used stringent controls to reduce the intra-experimental variation. This work is novel as there is no current literature describing the changes in looking at the cellular changes of R-loop mapping.

The claims are appropriately discussed when compared to earlier literature Hands et al., reference earlier literature when making claims, especially when referencing past literature in terms of p53 status and the effect of Aza and Dac when treating AML or MDS and as p53 is one of the most mutated genes in cancer this would be worth considering in the future.

I believe that clinicians will be most interested in this work or those who would be using Aza/Dac for treatment of AML. There are currently at least 6 hotspot mutations that are known in AML R175H, G245S, R248Q/W, R249S, R273H/S, and R282W. All these mutations are present in the DNA binding domain of p53, and either are partially or fully unfunctional p53 mutants. They may mirror the loss of p53 as shown by Hands et al., so looking into long term effects of Aza/Dac on those with these mutants would provide useful.

I believe the paper whilst confirming studies seen by others stands out in the field due to its involvement of p53 in the context of Aza/Dac treatment and the fact that stringent controls have been used when using the S9.6 antibody to prevent some of the intra-experimental variability normally observed when others use this. Due to its propensity to bind to dsRNA and DNA-RNA hybrids.

I believe the experimental data is sufficient quality to justify the conclusions present in this paper. The only figure which may need more clarity is Figure 2 F where the blots are not the cleanest, the loading control (b-actin) is fainter for the ctrl cells which may be why the phospho-p53 signal is also lower. There is also a vast amount of data in the supplemental figures to back up that shown in the main figures. The only other experiment I would suggest would be trying to see what happens to the epigenetic landscape and R-loop load when a mutant version of p53 is used instead of removing it entirely. 273 or 175H.

The comments are as follows:

1. HEK293 cells contain a bit of adenoviral genome that leads to expression of E1A/B proteins. It is widely known that this can also impact p53 and prevent p53 from regulating cell cycle or apoptosis (reviewed in <https://www.nature.com/articles/ncomms5767>). It is useful that the authors use the H1299 inducible line to verify results, but it would be good to highlight this point and speculate on how this may have affected results.
2. The impact of this manuscript would be raised if some experiments could be done to show how hotspot mutant versions of p53 affect R-loop load. Ideally in H1299 cells as mentioned above.
3. Figure 2F may need additional controls or quantification to prove p-p53 levels are lower.
4. Figure 5 could do with a control in which non-inducible or control-inducible H1299 cells that do not change p53 levels upon induction are treated with doxycycline to control for dox dependent effects.
5. Does p53 co-localise with R-loops?

Referee #2:

The manuscript « p53 status determines the epigenetic response to demethylating agents azacitidine and decitabine » by Langdale Hands and colleagues explored the contribution of p53 to gene expression modulation by demethylating agents.

The authors concluded that the response to aza could be dependent on the activation of p53 while that to dac would depend on chromatin modification and DNA damage. Although this study has undisputable merit and medical interest, notably, for the understanding of how combining aza-eprenetapopt (APR-246) may contribute to intercept the leukemic evolution of high-risk myelodysplastic syndromes, there are several concerns regarding the methodology and the conclusions.

1. Regarding the methodology, most of the techniques are state-of-the-art achievable techniques in cell lines. However, if DRIPc-seq is one the best manner to depict the R-loop landscape genome-wide starting from extracted DNA, the visualization of R-loops using immunofluorescence with S9.6 antibody in intact cells is rather inappropriate as the method unveils both RNA-DNA and RNA-RNA hybrids, the latter being much more frequent than RNA-DNA hybrids in a cell. Moreover, the appropriate control is not provided. The authors should have expressed by transfection RNaseH1 as a control of specificity.
2. In Fig. 1H, the authors considered the nuclear S9.6 signals. However, in Fig. S2B, the S9.6 signals were nuclear and cytosolic, questioning the specificity.
3. Epigenetic marks of closure (H3K27me3) and opening (H3K9me3) of the chromatin were analyzed by immunofluorescence (Fig. S1H/I), which prevented doing a global integration of epigenetic regulations at the level of DNA methylation and histone code. I would suggest investigating these marks using genome-wide methods like CUT&Tag.
4. The authors indicated that there is a huge difference in the intensity of DNA damage between aza versus dac treatments on the basis of excessive micronuclei in dac conditions. Because micronuclei may result from a lack of functional centromere or defects in proteins of the mitotic system resulting in chromosome segregation failure and because p53 and R-loop modulation may be implicated in the biological effects of aza or dac, the authors should provide with a deeper characterization of DNA damage by seeking double strand breaks or single-stranded DNA exposure. (Fig. S1D/F).
5. In DRIPc-seq experiments, the authors claimed that they were interested in looking at R-loops at promoters because the onset of R-loops at promoters positively regulated gene expression. Surprisingly, in their visualization of R-loop tracks (Fig. 2E; Fig. 3E) R-loops are more abundant at TTS in the chosen examples, which did not support the interpretation of the data. Is p53-target gene expression modulated in an R-loop dependent or independent manner? Is there recurrent differences in the expression level of genes that gained R-loop with aza and not with dac?
6. Could the authors explain why they refer to p53 activation by CBP/p300 acetylation and not study it? I would suggest moving this to the discussion.
7. To better investigate the discrepancy between p53 activation by aza (shown as an increased S15-pp53 expression) and the little changes in p53 target gene expression, the accessibility of chromatin at p53 binding site should be informative.

Referee #3:

In the manuscript entitled 'p53 status determines the epigenetic response to demethylating agents Azacitidine and Decitabine', the authors study the differential epigenetic effects of the hypomethylating agents (HMAs) Azacitidine (Aza) and decitabine (Dac). The HMAs have classically been thought to act through the same mechanism and are used interchangeably, despite mounting evidence that these drugs have multiple non-overlapping modes of action. In this study, the authors compare these drugs' effects on gene expression and methylation as well as analysis of the histone markers H3K9me3 and H3K27me3 to highlight non-overlapping epigenetic effects of the HMAs. Interestingly, the sometimes drastic differences in epigenetic effects of Aza vs Dec appear to be influenced by the induction of R-loops in a manner that is p53-dependent in the context of Aza.

Overall, this is a timely study of great interest to the scientific community as HMAs' clinical use has been given a new lease of life thanks in part to the effective combination of HMAs with venetoclax. Further support for the notion that Aza and Dec have non-overlapping effects, even, unexpectedly, at the epigenetic level, and a relationship between these effects and both R-loops and p53 status makes this an interesting study. However, I do have a small number of major concerns, mostly regarding the S9.6 antibody, that I think should be addressed (with achievable control experiments).

Major comments:

1. Overall, I find the manuscript ignores the mechanism by which Aza/Dac are thought to drive demethylation in the first place: the formation of DNMT1-DNA adducts, which are targeted by specific DNA repair machinery (see work from the Mailand, Stingele and Jackson labs) to degrade the DNMT1 and so deplete cells of DNMT1. Given that the epigenetic effects of Aza/Dac reported in this study are so different, I think it should be emphasized more that the mechanism of demethylation occurs should in principle be the same. This is highly relevant for the interpretation of results such as the DRIPc-seq, since it would suggest that a substantial subset of R-loops induced by one or the other drug do not arise from DNMT1-dependent demethylation.
2. As mentioned in the discussion, the sole use of the S9.6 antibody is not ideal for the analysis of R-loops. For example, S9.6 binds rRNA in nucleoli (Smolka et al, JCB 2021), and based on the DAPI signal in Fig 1H, it looks to me like many of the 'foci' counted are simply nucleoli that. Specifically, control approaches can be used to mitigate this, and I'd expect at least one of the following approaches to be taken. It's probably not practical to do all of these in every possible context, but at least some of the key observations should be supported with one of these controls.
 - a. Expression of GFP-RNASEH1 D210N mutant and repeat of this experiment for chromatin-associated GFP. This is a more robust readout of R-loops than S9.6.
 - b. Overexpression of WT RNASEH1 ahead of candidate R-loop-induced phenotypes such as Dac-induced micronuclei

c. pretreatment of wells/coverslips with RNase T1.

d. Costaining of S9.6 with a nucleolar marker such as NPM1 or nucleolin to allow separation of nucleolar 'foci' from non-nucleolar.

3. Details of the ApoTox assay are absent from the Methods. I also think the Results section would benefit from expansion of this to make the viability vs apoptosis vs cytotoxicity distinction more explicit. For example, I'm not sure why 10uM Dac seems to cause 5X greater cytotoxicity than NT at 2.5uM, but almost no impact on viability.

Minor comments:

1. The introduction states that Aza inhibits m5C in RNA. As alluded to in the Discussion, despite being a likely event this hasn't (to my knowledge) been shown directly - I'd suggest toning down that statement in the introduction.

2. As mentioned, S9.6 foci in Fig 1H are likely mostly nucleoli and might not actually be hugely relevant. It would be helpful to also include a zoomed-in image of just the S9.6 panel, perhaps with nuclei outlined rather than overlaid.

3. In general, it's sometimes unclear how many independent biological replicates are behind each quantification of IF data. For example, in S1E each dot represents a field, but are all the fields from one well/coverslip, or do they come from separate experiments? Such information should be added to each figure legend as it impacts on the robustness of the statistical tests.

4. It's surprising - and very interesting - that Aza/Dac cause such distinct effects on (e.g) expression, methylation R-loop distributions when assessed by DRIPc-seq. Given that the mechanism of demethylation (DNMT1 trapping and degradation) is thought to be the same for Aza/Dac, the findings could suggest:

a. 'Direct' demethylation effects through as-yet unknown mechanisms that are independent of the classical DNMT1 trapping route.

b. 'Indirect'/adaptive demethylation effects in response to non-overlapping impacts of Aza/Dac, for example in RNA for Aza, or through other DNMT1-independent effects of Dac (as per Carnie et al, EMBO J 2024 or Zhang et al EMBO Rep 2025) Mechanistically it's hard to see where these effects would come from, but it could be worth some speculation in the Discussion that these effects are not through the 'classical' route of DNMT1 degradation and might relate to the relatively long-term treatment times typically used in the study.

5. I think a simple 'sense-check' control experiment to see whether protein expression of RNASEH1 or members of the RNASEH2 complex would be helpful. This would probably rule out the (admittedly unlikely) possibility that R-loop induction after Aza/Dac is caused by formation of R-loops rather than impaired resolution.

6. Fig 2F: I'd suggest exchanging the p-p53 image for one with even b-actin signal if possible. Also, although this is with LiCOR and therefore not saturated, the p-53 blot would look overexposed to someone used to using ECL. I'd suggest showing a lower 'exposure' of the total p53 to avoid confusion.

7. Does p53 expression in the ip53 cells impact Aza/Dec cytotoxicity?

Referee #1:

>

> 1. Does this manuscript report a single key finding? YES

The loss of p53 changes the epigenetic landscape after Aza/Dac treatment.

> 2. Is the reported work of significance (YES), or does it describe a confirmatory finding or one that has already been documented using other methods or in other organisms etc (NO)? YES

Further expands how cells respond to changes in methylation but identifies a new possible biomarker for p53 status.

> 3. Is it of general interest to the molecular biology community? YES

Due to a deeper look into the further effects of Aza and Dac on p53 and the epigenetic landscape.

> 4. Is the single major finding robustly documented using independent lines of experimental evidence (YES), or is it really just a preliminary report requiring significant further data to become convincing, and thus more suited to a longer-format article (NO)?

> YES/NO

>

> Yes, No

>

The manuscript presented by Hands et al., uses Aza and Dac which are DNA methyltransferase inhibitors and looks at how, when given to cells at sub toxic concentrations changes in the methylation status of promoters occurs, this impacts the transcriptional profiles of treated cells. They also observed that R-loops were primarily formed in promoter regions when cells were treated with Aza vs the termination sites when treated with Dac. They also found out that knocking out p53 caused an increased amount of R-loop staining after Aza and Dac treatment when compared to the untreated controls. The authors used ap53 inducible cell line (H1299) to

show that when p53 was induced R-loop staining intensity was depleted even after treatment with Aza and Dac when compared to the non-induced cell line. The main takeaway was the fact that restoration of p53 is able to stabilise the epigenetic landscape and restore the R-loops. A strength of this manuscript lies in the discussion where a paragraph is set aside which mentions some limitations of the study, for example how variable the S9.6 antibody is, although this raises a question if another antibody should be used instead of the S9.6 antibody as this antibody has been shown to bind to dsRNA as well as the RNA-DNA hybrids.

The major claims are that p53 affects the formation of R-loops upon treatment and that p53 is a key player in how the cell responds to Aza and Dac treatment. The findings are significant as there has not been any previous work linking Aza and Dac treatment to p53 and its activation. Although previous work in the field has shown the dependence of p53 on R-loop formation in other areas such as during HPV

> (<https://eur01.safelinks.protection.outlook.com/?url=https%3A%2F%2Fwww>). Another major claim was that Aza and Dac treatment causes global remapping of R-loops, this work is significant because Aza and Dac are currently used in the treatment of AML and MDS so further research into how these drugs affect 'normal' cells in the body during treatment is important, to prevent any off-shoot effects occurring. Although the S9.6 antibody can be quite variable, as described earlier, Hands et al., used stringent controls to reduce the intra-experimental variation. This work is novel as there is no current literature describing the changes in looking at the cellular changes of R-loop mapping.

The claims are appropriately discussed when compared to earlier literature Hands et al., reference earlier literature when making claims, especially when referencing past literature in terms of p53 status and the effect of Aza and Dac when treating AML or MDS and as p53 is one of the most mutated genes in cancer this would be worth considering in the future

I believe that clinicians will be most interested in this work or those who would be using Aza/Dac for treatment of AML. There are currently at least 6 hotspot mutations that are known in AML R175H, G245S, R248Q/W, R249S, R273H/S, and R282W. All these mutations are present in the DNA binding domain of p53, and either are partially or fully unfunctional p53 mutants. They may mirror the loss of p53 as shown by Hands et al., so looking into long term effects of Aza/Dac on those with these mutants would provide useful. I believe the paper whilst confirming studies seen by others stands out in the field due to its involvement of p53 in the context of Aza/Dac treatment and the fact that stringent controls have been used when using the S9.6 antibody to prevent some of the intra-experimental variability normally observed when others use this. Due to its propensity to bind to dsRNA and DNA-RNA hybrids.

I believe the experimental data is sufficient quality to justify the conclusions present in this

paper. The only figure which may need more clarity is Figure 2 F where the blots are not the cleanest, the loading control (b-actin) is fainter for the ctrl cells which may be why the phospho-p53 signal is also lower. There is also a vast amount of data in the supplemental figures to back up that shown in the main figures. The only other experiment I would suggest would be trying to see what happens to the epigenetic landscape and R-loop load when a mutant version of p53 is used instead of removing it entirely. 273 or 175H.

We wanted to thank the reviewer for their detailed and considered comments assessing our manuscript. We have answered each of the comments below in a point by point manner, with changes to the text highlighted in red.

The comments are as follows:

> 1. HEK293 cells contain a bit of adenoviral genome that leads to expression of E1A/B proteins. It is widely known that this can also impact p53 and prevent p53 from regulating cell cycle or apoptosis (reviewed in <https://eur01.safelinks.protection.outlook.com/?url=https%3A%2F%2Fwww.> It is useful that the authors use the H1299 inducible line to verify results, but it would be good to highlight this point and speculate on how this may have affected results.

Thank you for raising this important point. We were conscious of relying on a single cell line to exclusively base our findings, especially so as the reviewer identifies the modifications regarding p53 biology and the HEK293 line. We have performed a series of additional experiments with the H1299 to further validate our findings, that are now included as Figure EV5 and Appendix Figure 1-2, which are discussed in further detail for the additional points raised. Further to this, we have now included in the discussion comments on these considerations, while also citing a study that has found similar changes following Dac exposure in a murine cancerous line, providing further support for these findings (I422-427).

> 2. The impact of this manuscript would be raised if some experiments could be done to show how hotspot mutant versions of p53 affect R-loop load. Ideally in H1299 cells as mentioned above.

We agree that this would be incredibly interesting to include mutants to further study the impact of impaired p53 function. Sufficiently interesting that we feel this would form the basis of an entirely independent study due to its potential size. Much work has been performed with approaches such as stable transfections of mutant p53 in the parent H1299 cell line. However we feel that for relevance alongside the data we have already generated, we would need to use the inducible system currently utilised for the rescue of WT p53. Technically, the modification of these H1299 cells to rescue with mutant p53, the validation that would be required with such novel lines, as we agree with the reviewer that multiple hot spot mutations would be required, and then the subsequent generation of the novel data, would be a substantial undertaking within any timeframe, far more so in the 3-month period available to us. Our analysis of the R-loop associated proteome also did not detect p53 directly associating with R-loops. As a result, we would likely need to include additional mutants outside of the DBD, expanding the scope of this

work further. For the HEK293^{-/-} cells, transfection with p53 expressing plasmids (wt or mutant) is incredibly challenging due to the toxic effects of p53 over-expression. We have therefore instead expanded our discussion to include a section discussing these mutants (I454-463) and further work to explore this novel role of p53.

> 3. Figure 2F may need additional controls or quantification to prove p-p53 levels are lower.

We thank the reviewer for raising this point. We have re-run the Westerns with a different p-p53 antibody and an increased n, to ensure clarity, and have updated the figures accordingly (Figure 2F, G). We continue to see Aza-specific accumulation of p-p53.

> 4. Figure 5 could do with a control in which non-inducible or control-inducible H1299 cells that do not change p53 levels upon induction are treated with doxycycline to control for dox dependent effects.

We agree that this control is needed. We have performed Aza and Dac exposures on Doxycycline exposed parent cells of the H1299 line, which do not possess inducible p53 activity, and performed S9.6 staining and quantification. These data can be found in Extended View Figure 5C-D and demonstrate an Aza specific accumulation of S9.6 staining. This result is different from our others, where the effect was more obviously observed in Dac treated cells (Figure 1I). However observation of the images (Figure EV5C) demonstrates substantial signal in Dac treated cells, which was not included within the nuclear specific quantification. Recent published work has highlighted the presence of cytosolic R-loops, which can be detected through immuno-staining, and are generally considered to represent a consequence of DNA damage (Crossley *et al*, 2023). We have discussed the presence of this cytosolic staining in the discussion.

Further analysis of DNA damage, as measured by γ H2Ax staining was performed on the H1299 line (solvent), the H1299 line (Dox) and the H1299 parent line (Dox), which can be found in Appendix Figure S2. We can observe Aza and Dac induced DNA damage in the H1299 line (solvent), a response rescued by the return of p53 (Dox). The Aza and Dac induced effects were maintained in the H1299 parent line, while γ H2Ax staining intensity was elevated for all three conditions. These experiments demonstrate that while Doxycycline does induce a measure of DNA damage, as identified by the reviewer likely to occur, we can also conclude that the changes observed in the Aza and Dac experiments represent effects specific to those drugs.

> 5. Does p53 co-localise with R-loops?

This is an interesting question and was part of our rationale for performing the R-loop based Mass spec experiments. We did not detect p53 in the pull downs for any of the conditions, indicating that in the HEK293 cells, p53 was not co-localising with R-loops. We have discussed this alongside considerations for future mutant experiments, as mutants outside of the DBD may be required to further elucidate the role of p53.

> -----

> Referee #2:

>

> The manuscript « p53 status determines the epigenetic response to demethylating agents azacitidine and decitabine » by Langdale Hands and colleagues explored the contribution of p53 to gene expression modulation by demethylating agents. The authors concluded that the response to aza could be dependent on the activation of p53 while that to dac would depend on chromatin modification and DNA damage. Although this study has undisputable merit and medical interest, notably, for the understanding of how combining aza-eprenetapopt (APR-246) may contribute to intercept the leukemic evolution of high-risk myelodysplastic syndromes, there are several concerns regarding the methodology and the conclusions.

We thank the reviewer for their time in reviewing our manuscript. We have provided a point-by-point response to their comments below, with changes to the manuscript highlighted in red.

1. Regarding the methodology, most of the techniques are state-of-the art achievable techniques in cell lines. However, if DRIPc-seq is one the best manner to depict the R-loop landscape genome-wide starting from extracted DNA, the visualization of R-loops using immunofluorescence with S9.6 antibody in intact cells is rather inappropriate as the method unveil both RNA-DNA and RNA-RNA hybrids, the latter being much more frequent than RNA-DNA hybrids in a cell. Moreover, the appropriate control is not provided. The authors should have expressed by transfection RNaseH1 as a control of specificity.

We thank the reviewer for their comments about the techniques we have used, and for raising this important point on the staining. Our experience with RNaseH1 over-expression results in substantial levels of cell death when used alongside these drugs, which we wished to avoid. We have therefore performed additional staining with the H1299 cell line using RNaseT1 treatment, at the suggestion of one of the other reviewers, to control for aberrant signals, which has been included in Figure EV5. While this approach will not resolve double stranded RNA structures, it will remove all single stranded RNA structures, including R-loops, and therefore we would expect any remaining signal to represent RNA-RNA structures that the antibody was detected. We see highly depleted levels of S9.6 staining in all conditions following RNaseT1 treatment, visualized and quantified in Figure EV5A-B. We hope this control is sufficient to provide confidence in the staining, however we have also included an expanded discussion point regarding the requirements for utilizing the S9.6-based approaches in a reproducible manner (I433-439).

2. In fig. 1H, the authors considered the nuclear S9.6 signals. However, in Fig. S2B, the S9.6 signals were nuclear and cytosolic, questioning the specificity.

The reviewer is correct that we restricted our analysis to nuclear S9.6 signals. To ensure confidence in the staining, we have performed additional S9.6 and Nucleolin staining on the H1299 cell line, with quantification, which can be found in Appendix Figure S1. In the presence

of p53 expression (Doxycycline), we see a highly enriched overlap between staining patterns for all three conditions (ctrl, Aza, Dac), as quantified by Pearson's and Mander's co-efficient (Appendix Figure S1B-E). In the absence of p53 expression (Solvent), this overlap was principally retained in ctrl and Aza treated cells, while a reduced correlation was observed in Dac treated cells (Appendix Figure S1A, C-E). Recent published work has highlighted the presence of cytosolic R-loops, which can be detected through immuno-staining (Crossley *et al*, 2023). While the sequencing approach we have utilized would not be able to confirm the presence of these R-loops, we would speculate that the significantly increased levels of DNA damage markers (Figure 1F-G, Figure EV3C-D, Appendix Figure S2A-D, Appendix Figure S2G-H) and micronuclei (Figure 1D-E, Figure EV3A-B, Appendix Figure S2E-F) which are most enriched in p53^{-/-} cells exposed to Dac, may indicate an accumulation of these cytosolic R-loops. We have added a discussion point on this subject (I358-361).

3. Epigenetic marks of closure (H3K27me3) and opening (H3K9me3) of the chromatin were analyzed by immunofluorescence (Fig. S1H/I), which prevented doing a global integration of epigenetic regulations at the level of DNA methylation and histone code. I would suggest investigating these marks using genome wide methods like CUT&Tag.

We agree that the interplay between different elements of the epigenome is an interesting and enduring question. We would highlight that a number of studies have previously explored the impact of Aza/Dac on elements of the epigenome, including RNA expression, DNA methylation and selected histone modifications, although we acknowledge that these studies were not performed with R-loops in mind. The authors of those studies noted that while broad changes in all of these marks were detected, changes correlated poorly, including those of DNA methylation and histone modifications. Authors in one of these studies (Epigenetics. 5:229–240.), performed with Aza on HEK293 cells (Dac was also used but the data was not shown), observed very modest overlap between H3K9me3 changes and H3K27me3 changes, with even less overlap for either mark with DNA methylation, concluding that the transcriptional changes they observed were only partially explainable by the epigenetic changes. We have therefore instead explored this interplay between the different marks in the discussion, and the differences that we observed in the responses to two drugs that share a common mechanism following the formation of DNMT1-DNA adducts (I391-411).

4. The authors indicated that there is a huge difference in the intensity of DNA damage between aza versus dac treatments on the basis of excessive micronuclei in dac conditions. Because micronuclei may result from a lack of functional centromere or defects in proteins of the mitotic system resulting in chromosome segregation failure and because p53 and R-loop modulation may be implicated in the biological effects of aza or dac, the authors should provide with a deeper characterization of DNA damage by seeking double strand breaks or single-stranded DNA exposure. (Fig. S1D/F).

We agree that further work here would provide greater confidence in the conclusions we have drawn. We have performed γ H2Ax staining with the H1299 and the H1299 parent cell lines, which can be found as a new figure (Appendix Figure S2). p53^{-/-} cells (Solvent) treated with Aza or Dac were found to possess significantly increased staining intensity over Ctrl cells (Appendix

Figure S2A, D). Rescue of p53 expression (Doxycycline) ablated this effect for both drugs (Appendix Figure S2B, D). Interestingly, in Doxycycline induced H1299 parent cells, which will not induce p53 rescue, γ H2Ax staining intensity was increased for all conditions, indicating Doxycycline induces DNA damage as would be expected, however the drug specific responses in the absence of p53 were maintained in these conditions (Appendix Figure S2C, D).

5. In DRIPc-seq experiments, the authors claimed that they were interested in looking at R-loops at promoters because the onset of R-loops at promoters positively regulated gene expression. Surprisingly, in their visualization of R-loop tracks (Fig. 2E; fig.3E) R-loops are more abundant at TTS in the chosen examples, which did not support the interpretation of the data. Is p53-target gene expression modulated in an R-loop dependent or independent manner? Is there recurrent differences in the expression level of genes that gained R-loop with aza and not with dac?

This is an important point, and one of the more unexpected elements of our data. We chosen to focus our peak tracks for the gene bodies of the genes and therefore the promoter regions of the genes are not shown, while the TTS are. This aligns with the data as the 3'UTR is the second most abundant R-loop location in the Aza and Dac groups. As the reviewer correctly raises, promoter R-loops are generally considered to be transcriptionally activating, however studies have demonstrated that at the individual gene level, this relationship is not so clearly defined (Stork *et al*, 2016; Wahba *et al*, 2016). We were surprised to find that almost none of the genes that gained promoter R-loops were significantly up-regulated (Figure EV2F), including the p53-target genes that we highlight. We performed the time-course with the expectation that the R-loop persisted beyond an earlier transcriptional response that may have been missed by the RNA-Seq experiments, however while some changes were detected (Figure EV2H), they were modest and transient. Unfortunately, we cannot reasonably perform DRIPc-Seq on all of the timepoints that we utilised for the expression time-course, which may shine light on the dynamics of these R-loops. However, we would highlight that our sequencing experiments were performed 48hrs after the initial exposures, and 24hrs after the final refresh, indicating persistent R-loop retention in the promoters of the genes, which is at odds with the generally accepted rapid turn-over of transcriptionally initiating promoter R-loops. Therefore, we have concluded that the R-loops that formed within these p53-target genes are not direct regulators of transcription and serve additional epigenetic functions. We have expanded our discussion to cover this point and the interplay between different elements of the epigenome. Regarding the second point, we did not detect any relationship between the expression level of a gene and its propensity with which to form an R-loop in either condition.

> 6. Could they authors explain why they refer to p53 activation byCBP/p300 acetylation and not study it? I would suggest moving this to the discussion.

Thank you for this suggestion. We have moved this sentence to the discussion alongside further points regarding the Aza specific activation of p53.

> 7. To better investigating the discrepancy between p53 activation by aza (shown as an increased S15-pp53 expression) and the little changes in p-53 target gene expression, the

accessibility of chromatin at p53 binding site should be informative.

To provide greater clarity, we have repeated the blotting experiments to demonstrate pp53 accumulation. This has now been included in the updated Figure 2F, alongside total p53 and B-Actin loading control and quantification. We continue to see Aza-specific enrichment of pp53. Regarding the accessibility of chromatin, treatment of Dac (in MCF-7 cells) followed by ATAC-Seq and p53-ChIP-Seq was found to only modestly alter accessibility in p53 binding sites (Hafner *et al*, 2020), with the authors concluding that the global p53 DNA binding profile was largely unchanged. As the reviewer notes, we only saw very modest changes in p53-target gene expression following either Aza or Dac treatment at any point in the time-course study, with no changes detected in the RNA-Seq, while newly formed R-loops sites in p53-target genes were not found to clearly overlap with p53 binding sites. We therefore felt that performing ATAC-Seq, especially given the high levels of variability that were detected in some of the genome-wide approaches that we performed for Dac, was unlikely to provide novel insights into the response of these drugs.

> -----

> Referee #3:

>

> In the manuscript entitled 'p53 status determines the epigenetic response to demethylating agents Azacitidine and Decitabine', the authors study the differential epigenetic effects of the hypomethylating agents (HMAs) Azacitidine (Aza) and decitabine (Dac). The HMAs have classically been thought to act through the same mechanism and are used interchangeably, despite mounting evidence that these drugs have multiple non-overlapping modes of action. In this study, the authors compare these drugs' effects on gene expression and methylation as well as analysis of the histone markers H3K9me3 and H3K27me3 to highlight non-overlapping epigenetic effects of the HMAs. Interestingly, the sometimes drastic differences in epigenetic effects of Aza vs Dec appear to be influenced by the induction of R-loops in a manner that is p53-dependent in the context of Aza.

Overall, this is a timely study of great interest to the scientific community as HMAs' clinical use has been given a new lease of life thanks in part to the effective combination of HMAs with venetoclax. Further support for the notion that Aza and Dec have non-overlapping effects, even, unexpectedly, at the epigenetic level, and a relationship between these effects and both R-loops and p53 status makes this an interesting study. However, I do have a small number of major concerns, mostly regarding the S9.6 antibody, that I think should be addressed (with achievable control experiments).

We wanted to thank the reviewer for their time taken in reviewing our manuscript. We have addressed their comments in a point-by-point basis below, which we feel has helped to further support the conclusions that we have drawn. Changes to the text are highlighted in red.

> Major comments:

>

> 1. Overall, I find the manuscript ignores the mechanism by which Aza/Dac are thought to drive demethylation in the first place: the formation of DNMT1-DNA adducts, which are targeted by specific DNA repair machinery (see work from the Mailand, Stingele and Jackson labs) to degrade the DNMT1 and so deplete cells of DNMT1. Given that the epigenetic effects of Aza/Dac reported in this study are so different, I think it should be emphasised more that the mechanism of demethylation occurs should in principle be the same. This is highly relevant for the interpretation of results such as the DRIPc-seq, since it would suggest that a substantial subset of R-loops induced by one or the other drug do not arise from DNMT1-dependent demethylation.

Thank you for this suggestion, and for raising this important point. Many studies that have explored the impact of these drugs on the epigenome beyond the context of DNA methylation have generally been performed with either Aza or Dac being studied in isolation, but rarely together. While individually, our data for each drug aligns closely with many of these studies, our use of both drugs in the same context highlights the number of differences which would not be predicted. One of the most surprising elements has been the observation that drugs which should principally be acting through DNMT1-DNA adducts, as the reviewer raises, have produced such variable responses within the broader context of the epigenome. We have added a sentence in the introduction (179-84) to highlight this and have speculated in the discussion about the surprising differences in the broad epigenetic consequences of these drugs, which we speculate occur due to the Aza-specific activation of p53 inducing a programmed response to the drug, while we believe the Dac-specific changes represent a consequential response of the drug.

> 2. As mentioned in the discussion, the sole use of the S9.6 antibody is not ideal for the analysis of R-loops. For example, S9.6 binds rRNA in nucleoli (Smolka et al, JCB 2021), and based on the DAPI signal in Fig 1H, it looks to me like many of the 'foci' counted are simply nucleoli that. Specifically, control approaches can be used to mitigate this, and I'd expect at least one of the following approaches to be taken. It's probably not practical to do all of these in every possible context, but at least some of the key observations should be supported with one of these controls.

> a. Expression of GFP-RNASEH1 D210N mutant and repeat of this experiment for chromatin-associated GFP. This is a more robust readout of R-loops than S9.6.

b. Overexpression of WT RNASEH1 ahead of candidate R-loop-induced phenotypes such as Dac-induced micronuclei c. pretreatment of wells/coverslips with RNase T1.

d. Costaining of S9.6 with a nucleolar marker such as NPM1 or nucleolin to allow separation of nucleolar 'foci' from non-nucleolar.

Thank you for these helpful suggestions, and we agree that adding controls to provide confidence in the staining will benefit the manuscript. We have opted for additional RNase T1 experiments and Nucleolin co-staining experiments as suggested by the reviewer, which we have performed on the H1299 cell line, as requested by one of the other reviewers. These data can be found in Extended View Figure 5A-B (RNase T1) and Appendix Figure S1 (Nucleolin/S9.6 co-staining). Treatment with RNase T1 caused a highly significant depletion of S9.6 staining in all conditions (Ctrl, Aza and Dac). Nucleolin staining in the presence of p53 in the H1299 cell line (Dox) was found to correlate highly with S9.6 staining, as quantified by Pearson's and Mander's coefficient (Appendix Figure S1B-E). In the absence of p53 (Solvent), this correlation was principally retained for Ctrl and Aza treated cells, while a reduced correlation was observed in Dac treated cells (Appendix Figure S1A, C-E). Further to this, we have included an expanded discussion point regarding the appropriate controls that are required when using S9.6 based approaches (I433-439).

> 3. Details of the ApoTox assay are absent from the Methods. I also think the Results section would benefit from expansion of this to make the viability vs apoptosis vs cytotoxicity distinction more explicit. For example, I'm not sure why 10uM Dac seems to cause 5X greater cytotoxicity than NT at 2.5uM, but almost no impact on viability.

We apologize for the lack of clarity on these experiments. We have updated the methods to provide further details, and have included further narrative in the results section to describe these experiments more completely. This assay acts as an end point assessment, and therefore provides a snapshot at that particular moment in time, which can preclude the observation of earlier markers of toxicity. We therefore ensured that our dose determination was based off all three measures and checked alongside previously published reports to utilise doses that minimised cell death while ensuring DNA de-methylation was still achieved.

> Minor comments:

> 1. The introduction states that Aza inhibits m5C in RNA. As alluded to in the Discussion, despite being a likely event this hasn't (to my knowledge) been shown directly - I'd suggest toning down that statement in the introduction.

Thank you for this suggestion, the reviewer is correct that no studies had in fact directly shown changes in m5C levels in RNA following Aza treatment when we originally prepared this paper. Somewhat fortuitously, a very recently published study (Molecular Carcinogenesis, 2025; 64:502–512) has now demonstrated changes in RNA m5C levels in patients following Aza treatment. We have now included this citation in the introduction alongside this statement, which we hope the reviewer agrees can now be justified.

> 2. As mentioned, S9.6 foci in Fig 1H are likely mostly nucleoli and might not actually be hugely relevant. It would be helpful to also include a zoomed-in image of just the S9.6 panel, perhaps with nuclei outlined rather than overlaid.

We hope that our answer above regarding the additional nucleolin and S9.6 co-staining has provided sufficient confidence in the staining.

> 3. In general, it's sometimes unclear how many independent biological replicates are behind each quantification of IF data. For example, in S1E each dot represents a field, but are all the fields from one well/coverslip, or do they come from separate experiments? Such information should be added to each figure legend as it impacts on the robustness of the statistical tests.

We apologize that this information was not made clearer in the initial submission. We have added this information to each legend now.

> 4. It's surprising - and very interesting - that Aza/Dac cause such distinct effects on (e.g) expression, methylation R-loop distributions when assessed by DRIPc-seq. Given that the mechanism of demethylation (DNMT1 trapping and degradation) is thought to be the same for Aza/Dac, the findings could suggest: a. 'Direct' demethylation effects through as-yet unknown mechanisms that are independent of the classical DNMT1 trapping route. b. 'Indirect'/adaptive demethylation effects in response to non-overlapping impacts of Aza/Dac, for example in RNA for Aza, or through other DNMT1-independent effects of Dac (as per Carnie et al, EMBO J 2024 or Zhang et al EMBO Rep 2025) Mechanistically it's hard to see where these effects would come from, but it could be worth some speculation in the Discussion that these effects are not through the 'classical' route of DNMT1 degradation and might relate to the relatively long-term treatment times typically used in the study.

We agree with the reviewer that these observations were both surprising and interesting. Our analysis of genome-wide DNA methylation did demonstrate differential depletion of DNA methylation levels, in this case, we observed a generally greater depletion from Aza. As has been observed previously, the level of induced hypo-methylation of these drugs is highly context specific. Our data has not shone any light on the possibility of novel direct demethylation effects, which as the reviewer identifies, are challenging to imagine where they may come from. The potential for indirect effects remains in our view the more likely route for such differences in the data. We speculate that the drug specific formation of R-loops can be included as such an indirect effect. Following on from our response to major point 1, we have added in some further discussion on these differences and highlighted the role R-loop formation may be playing in modulating some of these differences.

> 5. I think a simple 'sense-check' control experiment to see whether protein expression of RNASEH1 or members of the RNASEH2 complex would be helpful. This would probably rule out the (admittedly unlikely) possibility that R-loop induction after Aza/Dac is caused by formation of R-loops rather than impaired resolution.

We include RNASEH1 blotting below from HEK293 cells to demonstrate protein expression is unchanged in either drug treatment. We would be happy to include this in the manuscript, however we felt initially that there was not an obvious place for it.

> 6. Fig 2F: I'd suggest exchanging the p-p53 image for one with even b-actin signal if possible. Also, although this is with LiCOR and therefore not saturated, the p-53 blot would look overexposed to someone used to using ECL. I'd suggest showing a lower 'exposure' of the total p53 to avoid confusion.

Thank you for this suggestion. We have performed additional blots, with an increased n. These have been quantified and the figure updated accordingly, with the Aza-specific activation of p-p53 still observed.

> 7. Does p53 expression in the ip53 cells impact Aza/Dec cytotoxicity?

This is an interesting question. Our newly generated \square H2AX, micro-nuclei and \square tubulin imaging data (Appendix Figure S2) demonstrated that p53 expression/rescue reduced DNA damage, and that this was sufficient to rescue both the drug specific effects, but also the effect of the DOX. While this in itself is not a direct measure of cytotoxicity, we feel it is unlikely that the expression of p53 in this system would be inducing cytotoxicity.

Dear Dr. Van de Pette,

Thank you for the submission of your revised manuscript to our editorial offices. I have now received the reports from the three referees that I asked to re-evaluate the study, you will find below. As you will see, the referees now fully support the publication of your study in EMBO reports. Referee #2 has a suggestion to improve the manuscript, I ask you to address in a final revised manuscript. Please also provide a final p-b-p-response to this point and to the editorial requests below.

Editorial requests:

- Please provide the abstract written in present tense throughout.
- Please remove the table of contents from the manuscript text file.
- The nomenclature of the EV figures is not correct in the legends. Please use 'Figure EVx'.
- Please add the database name for the datasets GSE243785, GSE243786 and GSE243784 to the data availability statement, add a brief explanation what each dataset contains and add direct links to the datasets.
- Please check again that the number "n" for how many independent experiments were performed, their nature (biological versus technical replicates), the bars and error bars (e.g. SEM, SD) and the test used to calculate p-values is indicated in the respective figure legends. Please also check that all the p-values are explained in the legend, and that these fit to those shown in the figure. Please provide statistical testing where applicable. Please avoid the phrase 'independent experiment' but clearly state if these were biological or technical replicates. Please also indicate (e.g. with n.s.) if testing was performed, but the differences are not significant. In case n=2, please show the data as separate datapoints without error bars and statistics (please check for panels 1I, and EV5B/D). See also:
<http://www.embopress.org/page/journal/14693178/authorguide#statisticalanalysis>
- If n<5, please show single datapoints for diagrams. Moreover:
 - Please note that the exact p values are not provided in the legends of figures 1B, C, I; 2F, 3A, 5E, H, K; EV1 B, E, H, I; EV2 H, EV3 B, E, F; EV5 B, D
 - Please indicate the statistical test used for data analysis in the legends of figures 2D, 4C
 - Please note that the box plots need to be defined in terms of minima, maxima, centre, bounds of box and whiskers, and percentile in the legends of figures 1C, EV1 B, E, H, I; EV3 B, E, F
- Appendix Table S1 is a dataset. Please remove the table from the Appendix file and upload the original Excel file as a dataset file (named 'Dataset EV1'). Please add a legend on the first TAB of the Excel file. Finally, please update all callouts.
- Please remove Appendix Table S2 from the Appendix. Please add the primer information directly to the Reagents & Tools table and update the callouts. Please also remove the instructions from the Reagents & Tools table.
- Please update the TOC of the Appendix and add page numbers. Moreover, the Appendix items need to be named 'Appendix Figure Sx' also in the TOC. Please do that.
- During our figure integrity analysis, we found these issues:
 1. The top and bottom images in Fig. 3B (p53 KO Ctrl and p53 KO DAC) in the second column (R-loops) appear to be identical. Please check.
 2. The central part of the second panel (AZA) of Figure EV1D seems to overlap with a part of the central panel (AZA) in the second column (Merge with DAPI) of Figure EV1H. Please check. If this re-use is intentional, please clearly state this in the legends for these panels.
 3. The central part of the third (bottom) panel (DAC) of the first column (DAPI) of Appendix Fig S1B seems to overlap with the central part of the second panel (ip53 H1299 +Dox) of the first column (Ctrl) of Appendix Fig S2F. Please check. If this re-use is intentional, please clearly state this in the legends for these panels.
- Thanks for providing the source data. Please upload this as one folder per main figure, grouping together all the files for this figure (and ZIPed together), and as one folder for the EV figures containing separate folders for each EV figure. Or it is presently unclear what source data has been provided for Fig. 4 and Fig. EV4 (as the source data checklist indicates that no SD has been uploaded for Fig. 4 - presenting DRIP-Mass Spectrometry data?).
- I can't find indications that the source data for the DRIP-Mass Spectrometry (used for Fig. 4?) has been deposited (GSE243785, GSE243786 and GSE243784 seem not to contain this). Please add this to the Data availability section.

In addition, I would need from you uploaded separately:

Best,

Referee #1:

I feel the authors have sufficiently addressed my comments.

Referee #2:

I thank the authors for addressing all my comments by providing elements to the discussion and additional experiments. They assessed the specificity of immunofluorescence labelling of R-loops by S9.6 antibody with RNaseT1 pretreatment, R-loop localization to the nucleolus by co-labelling with antibody to nucleolin and DNA damage with γ H2AX antibody. I kindly suggest to dampen the interpretation of R-loop/nucleolin co-labelling that shows little or no differences between aza and dac and to increase the number of replicates and images for DNA damage analysis to consolidate the results.

Referee #3:

The authors have addressed all of my comments/suggestions and should be congratulated on a thorough and interesting study.

Dear Dr. Breiling,

Thank you for yours and the reviewer's comments. We have updated the manuscript as requested, with each point outlined below.

Editorial requests:

- Please provide the abstract written in present tense throughout. **Completed and highlighted changes in red**

- Please remove the table of contents from the manuscript text file. **Completed**

- The nomenclature of the EV figures is not correct in the legends. Please use 'Figure EVx'. **Updated in legends to correct nomenclature**

- Please add the database name for the datasets GSE243785, GSE243786 and GSE243784 to the data availability statement, add a brief explanation what each dataset contains and add direct links to the datasets. **Name, explanation and links added. DRIP-Mass also added.**

- Please check again that the number "n" for how many independent experiments were performed, their nature (biological versus technical replicates), the bars and error bars (e.g. SEM, SD) and the test used to calculate p-values is indicated in the respective figure legends. Please also check that all the p-values are explained in the legend, and that these fit to those shown in the figure. Please provide statistical testing where applicable. Please avoid the phrase 'independent experiment' but clearly state if these were biological or technical replicates. Please also indicate (e.g. with n.s.) if testing was performed, but the differences are not significant. In case n=2, please show the data as separate datapoints without error bars and statistics (please check for panels 1I, and EV5B/D). See also:

<http://www.embopress.org/page/journal/14693178/authorguide#statisticalanalysis>

Legends have been updated as requested. Use of "independent experiment" have been updated as requested.

If n<5, please show single datapoints for diagrams. Moreover:

- Please note that the exact p values are not provided in the legends of figures 1B, C, I; 2F, 3A,

5E, H, K; EV1 B, E, H, I; EV2 H, EV3 B, E, F; EV5 B, D. **These have been added to the legends**

- Please indicate the statistical test used for data analysis in the legends of figures 2D, 4C. **This information has now been included.**

- Appendix Table S1 is a dataset. Please remove the table from the Appendix file and upload the original Excel file as a dataset file (named 'Dataset EV1'). Please add a legend on the first TAB of the Excel file. Finally, please update all callouts. **Done**

- Please remove Appendix Table S2 from the Appendix. Please add the primer information directly to the Reagents & Tools table and update the callouts. Please also remove the instructions from the Reagents & Tools table. **Done**

- Please update the TOC of the Appendix and add page numbers. Moreover, the Appendix items need to be named 'Appendix Figure Sx' also in the TOC. Please do that. **Done**

- During our figure integrity analysis, we found these issues:

1. The top and bottom images in Fig. 3B (p53 KO Ctrl and p53 KO DAC) in the second column (R-loops) appear to be identical. Please check. **We have updated this figure, we apologize for this and are not sure how this duplication has appeared.**

2. The central part of the second panel (AZA) of Figure EV1D seems to overlap with a part of the central panel (AZA) in the second column (Merge with DAPI) of Figure EV1H. Please check. If this re-use is intentional, please clearly state this in the legends for these panels. **We have updated the legends to clarify this**

3. The central part of the third (bottom) panel (DAC) of the first column (DAPI) of Appendix Fig S1B seems to overlap with the central part of the second panel (ip53 H1299 +Dox) of the first column (Ctrl) of Appendix Fig S2F. Please check. If this re-use is intentional, please clearly state this in the legends for these panels. **We have updated the legends to clarify this**

- Thanks for providing the source data. Please upload this as one folder per main figure, grouping together all the files for this figure (and ZIPed together), and as one folder for the EV figures containing separate folders for each EV figure. Or it is presently unclear what source data has been provided for Fig. 4 and Fig. EV4 (as the source data checklist indicates that no SD has been uploaded for Fig. 4 - presenting DRIP-Mass Spectrometry data?). **We have updated these files and uploaded. Source data information for the DRIP-Mass is now included in the data availability statement**

- I can't find indications that the source data for the DRIP-Mass Spectrometry (used for Fig. 4?) has been deposited (GSE243785, GSE243786 and GSE243784 seem not to contain this). Please add this to the Data availability section. **Done**

In addition, I would need from you uploaded separately:

- a short, two-sentence summary of the manuscript (not more than 35 words).

Azacitidine and Decitabine induce divergent changes to R-loops and the wider epigenome, subject to p53 mediated control. Removal of p53 ablates these effects, while the damaging impact of p53 removal on the epigenome is reversible.

- two to four short (!) bullet points highlighting the key findings of your study (two lines each).

Azacitidine and Decitabine modulate the wider epigenome in divergent manners, in part through changes to R-loop patterns.

Azacitidine activates p53 and requires its activation for an appropriate epigenetic response

Decitabine does not activate p53, but induces substantial DNA damage in its absence

The epigenetic consequences of p53 removal are reversible.

- a schematic summary figure as separate file that provides a sketch of the major findings (not a data image) in jpeg or tiff format (with the exact width of 550 pixels and a height of not more than 400 pixels) that can be used as a visual synopsis on our website. **We have uploaded a figure**

Please use this link to submit your revision: <https://embor.msubmit.net/cgi-bin/main.plex>

Best,

Referee #1:

I feel the authors have sufficiently addressed my comments.

Thank you to the reviewer for the time they have taken in both rounds of peer review.

Referee #2:

I thank the authors for addressing all my comments by providing elements to the discussion and additional experiments. They assessed the specificity of immunofluorescence labelling of R-loops by S9.6 antibody with RNaseT1 pretreatment, R-loop localization to the nucleolus by co-labelling with antibody to nucleolin and DNA damage with γ H2AX antibody. I kindly suggest to dampen the interpretation of R-loop/nucleolin co-labelling that shows little or no differences between aza and dac and to increase the number of replicates and images for DNA damage analysis to consolidate the results.

Thank you to the reviewer for their comments across both rounds of peer review. We have modified the language in the manuscript at their suggestion (highlighted in red), while we have increased the n of the DNA damage analysis.

Referee #3:

The authors have addressed all of my comments/suggestions and should be congratulated on a thorough and interesting study.

Thank you to the reviewer for the time they have taken in both rounds of peer review.

Dr. Mathew Van de Pette
University of Cambridge
MRC Toxicology Unit
Gleeson Building
Tennis Court Road
Cambridge, Cambridgeshire CB2 1QR
United Kingdom

Dear Dr. Van de Pette,

I am very pleased to accept your manuscript for publication in the next available issue of EMBO reports. Thank you for your contribution to our journal.

You may qualify for financial assistance for your publication charges - either via a Springer Nature fully open access agreement or an EMBO initiative. Check your eligibility: <https://link.springer.com/journal/44319/how-to-publish-with-us>

Yours sincerely,

>>> Please note that it is EMBO Reports policy for the transcript of the editorial process (containing referee reports and your response letter) to be published as an online supplement to each paper. If you do NOT want this, you will need to inform the Editorial Office via email immediately. More information is available here: <https://link.springer.com/partners/embo-press/editorial-policies#Peer%20review>